METHODS AND RESOURCES

# A sensitive and affordable multiplex RT-qPCR assay for SARS-CoV-2 detection

**Martin A. M. Reijns**[1]*, **Louise Thompson**[2], **Juan Carlos Acosta**[3], **Holly A. Black**[2,4], **Francisco J. Sanchez-Luque**[1,5], **Austin Diamond**[2], **David A. Parry**[1], **Alison Daniels**[6], **Marie O'Shea**[7], **Carolina Uggenti**[4], **Maria C. Sanchez**[6], **Alan O'Callaghan**[1], **Michelle L. L. McNab**[6], **Martyna Adamowicz**[4], **Elias T. Friman**[1], **Toby Hurd**[1], **Edward J. Jarman**[1], **Frederic Li Mow Chee**[3], **Jacqueline K. Rainger**[1], **Marion Walker**[3], **Camilla Drake**[1], **Dasa Longman**[1], **Christine Mordstein**[1,8], **Sophie J. Warlow**[4], **Stewart McKay**[1], **Louise Slater**[2], **Morad Ansari**[2], **Ian P. M. Tomlinson**[3], **David Moore**[2], **Nadine Wilkinson**[9], **Jill Shepherd**[9], **Kate Templeton**[9], **Ingolfur Johannessen**[9], **Christine Tait-Burkard**[7], **Jürgen G. Haas**[6], **Nick Gilbert**[1], **Ian R. Adams**[1], **Andrew P. Jackson**[1]

**1** MRC Human Genetics Unit, MRC Institute of Genetics and Molecular Medicine, The University of Edinburgh, Edinburgh, United Kingdom, **2** The South East of Scotland Clinical Genetic Service, Western General Hospital, NHS Lothian, Edinburgh, United Kingdom, **3** Cancer Research UK Edinburgh Centre, MRC Institute of Genetics and Molecular Medicine, The University of Edinburgh, Edinburgh, United Kingdom, **4** Centre for Genomic & Experimental Medicine, MRC Institute of Genetics and Molecular Medicine, The University of Edinburgh, Edinburgh, United Kingdom, **5** Centre Pfizer-University of Granada-Andalusian Government for Genomics and Oncological Research (Genyo), Granada, Spain, **6** Division of Infection Medicine, Edinburgh Medical School, The University of Edinburgh, Edinburgh, United Kingdom, **7** The Roslin Institute and Royal (Dick) School of Veterinary Studies, The University of Edinburgh, Edinburgh, United Kingdom, **8** The Milner Centre for Evolution, Department of Biology and Biochemistry, University of Bath, Bath, United Kingdom, **9** Medical Microbiology and Virology Service, Royal Infirmary of Edinburgh, NHS Lothian, Edinburgh, United Kingdom

* martin.reijns@igmm.ed.ac.uk

**Data Availability Statement:** All relevant data are within the paper and its Supporting Information files.

## Abstract

With the ongoing COVID-19 (Coronavirus Disease 2019) pandemic, caused by the novel coronavirus SARS-CoV-2 (Severe Acute Respiratory Syndrome Coronavirus 2), there is a need for sensitive, specific, and affordable diagnostic tests to identify infected individuals, not all of whom are symptomatic. The most sensitive test involves the detection of viral RNA using RT-qPCR (quantitative reverse transcription PCR), with many commercial kits now available for this purpose. However, these are expensive, and supply of such kits in sufficient numbers cannot always be guaranteed. We therefore developed a multiplex assay using well-established SARS-CoV-2 targets alongside a human cellular control (*RPP30*) and a viral spike-in control (Phocine Herpes Virus 1 [PhHV-1]), which monitor sample quality and nucleic acid extraction efficiency, respectively. Here, we establish that this test performs as well as widely used commercial assays, but at substantially reduced cost. Furthermore, we demonstrate >1,000-fold variability in material routinely collected by combined nose and throat swabbing and establish a statistically significant correlation between the detected level of human and SARS-CoV-2 nucleic acids. The inclusion of the human control probe in our assay therefore provides a quantitative measure of sample quality that could help reduce false-negative rates. We demonstrate the feasibility of establishing a robust RT-

**Funding:** This work was supported by a unit programme grant from the Medical Research Council (MRC) UK to the MRC Human Genetics Unit; Biotechnology and Biological Sciences Research Council (BBSRC) Institute Strategic Programme grant funding to The Roslin Institute (BBS/E/D/20002172, BBS/E/D/20002173); and Medical Research Council UK funding to APJ (MC_UU_00007/5). https://mrc.ukri.org/ and https://bbsrc.ukri.org/. National Health Service (NHS) National Services Scotland also provided support for the project, through funding the NHS-led University of Edinburgh/NHS Lothian COVID-19 testing centre at the MRC Institute of Genetics and Molecular Medicine. https://nhsnss.org/. The funders had no role in study design, data collection and analysis, decision to publish, or preparation of the manuscript.

**Competing interests:** The authors have declared that no competing interests exist.

**Abbreviations:** CDC, Centers for Disease Control and Prevention; CFR, CAL Fluor Red; COVID-19, Coronavirus Disease 2019; Cq, cycle quantification; Ct, cycle threshold; CVOP, Coronavirus Outbreak Preparedness; ddPCR, droplet digital PCR; E, envelope; FAM, fluorescein amidite; IDT, Integrated DNA Technologies; IUPAC, International Union of Pure and Applied Chemistry; IVT, in vitro transcribed; MSA, multiple sequence alignment; N, nucleocapsid; NTS, nose and throat swabs; ORF, open reading frame; PhHV, Phocine Herpes Virus; QCMD, Quality Control for Molecular Diagnostics; RdRp, RNA-dependent RNA polymerase; RNase, ribonuclease; RT-qPCR, quantitative reverse transcription PCR; SARS-CoV-2, Severe Acute Respiratory Syndrome Coronavirus 2; VTM, viral transport medium; WHO, World Health Organization.

qPCR assay at approximately 10% of the cost of equivalent commercial assays, which could benefit low-resource environments and make high-volume testing affordable.

## Introduction

The COVID-19 (Coronavirus Disease 2019) pandemic, caused by the novel coronavirus SARS-CoV-2 (Severe Acute Respiratory Syndrome Coronavirus 2) [1], originated in Wuhan (China) in December 2019 and rapidly spread across the globe, resulting in substantial mortality [2,3] and widespread economic damage. Until a vaccine becomes available, public health strategies centered on reducing the rate of transmission are crucial to mitigating the epidemic, for which effective and affordable testing strategies to enable widespread population surveillance are essential. The most sensitive test to diagnose infected individuals involves the detection of SARS-CoV-2 viral RNA using RT-qPCR (quantitative reverse transcription PCR), most commonly using samples collected with nasopharyngeal (combined nose and throat) swabs (NTS), although there is increasing evidence that the use of saliva may be a valid alternative [4–7]. Many commercial kits are now available, most of which employ multiplex RT-qPCR, detecting 2 or 3 different SARS-CoV-2 targets, and generally include an internal control to show successful nucleic acid extraction. However, such kits are often costly, and their supply in sufficient numbers cannot always be guaranteed. We therefore developed a similar multiplex assay using well-established SARS-CoV-2 targets and internal controls, which can be carried out at a significantly lower cost and provides more flexibility to ensure resilience against potential shortages in reagent supplies.

Our assay makes use of the Takara One Step PrimeScript III RT-qPCR kit (Takara Bio, Shiga, Japan). This reagent was used in the first high-profile publication to describe SARS-CoV-2 [1], and it has since been shown to outperform a number of similar reagents [8]. Before commercial COVID-19 assays were available, various in-house assays were published on the WHO website [9]. Based on the data available at the time (March 2020), we decided to focus our initial efforts on targeting the following SARS-CoV-2 genes (Fig 1A, S1 File; [10–12]): envelope (E), RNA-dependent RNA polymerase (RdRp), and nucleocapsid (N). Corman and colleagues proposed the E gene as a useful target for first-line screening, with the RdRp gene suggested as a good target for confirmatory/discriminatory assays [13,14]. The N gene was central to the United States of America Centers for Disease Control and Prevention (CDC) in vitro diagnostics emergency protocol, with 3 different primer/probe sets (N1, N2, and N3) used against different portions of this viral gene [9]. The CDC protocol also included a probe against human *RPP30*, a single-copy gene encoding the protein subunit p30 of the ribonuclease (RNase) P particle, to ensure the presence of a sufficient number of cells in patient samples and successful isolation of intact nucleic acids.

In early versions of these protocols, all probes were labelled with fluorescein amidite (FAM), and separate reactions were therefore needed to detect each target. To increase efficiency, we developed a multiplex assay using 4 different fluorescent labels (FAM, HEX, CAL Fluor Red (CFR) 610, and Quasar 670) for each of the probes, allowing their detection in a single reaction. In the final version of our assay, we use previously described primers and probes against the well-established SARS-CoV-2 E and N gene (N1 and N2) targets, as well as a human cellular "sample quality control" and a viral spike-in "extraction control" (Fig 1B and 1C): human *RPP30* and Phocine Herpes Virus 1 (PhHV-1, hereafter referred to as PhHV). The rationale behind the human cellular control is that a considerable number of patients with clinical and radiological signs of COVID-19 are PCR–negative, and poor quality of swab

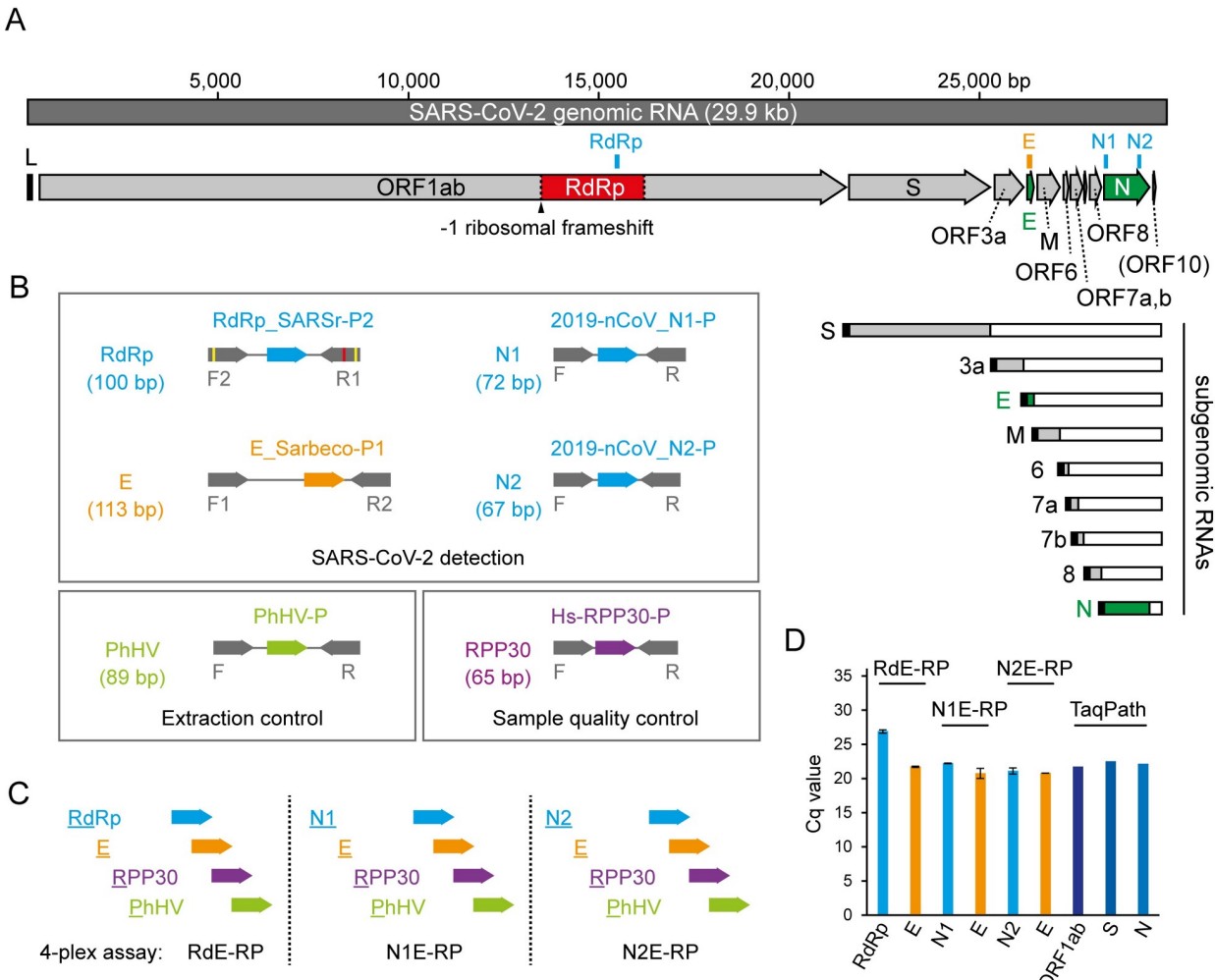

**Fig 1. Primers and probes used in our multiplex qRT-PCR assays detect SARS-CoV-2 RNA.** (A) Location of qRT-PCR amplicons on the SARS-CoV-2 genome. ORFs and qRT-PCR target sites (orange and blue) in RdRp, E, and N gene indicated. After a −1 ribosomal frameshift (arrowhead) on ORF1ab of the genomic RNA, the pp1ab polypeptide is formed, and RdRp/nsp12 is released by proteolytic cleavage (dotted lines). Genes 3′ of ORF1ab on the positive-sense genome (including E and N) are transcribed (from negative-sense RNA) as subgenomic RNAs and include a short leader sequence (L, black box) at their 5′ ends [10]. (B) qRT-PCR primers (grey) and probes (colour) used in this study. PCR product sizes are indicated between brackets. Yellow and red lines in RdRp primers indicate degenerate nucleotides and a mismatch, respectively (as present in the original design [14]). To detect RdRp, we used the RdRp_SARSr-P2 probe, which detects 2019-nCoV/SARS-CoV-2 and not SARS-CoV. The N1 and N2 assays are similarly specific to SARS-CoV-2, whereas the E gene assay also detects SARS-CoV [9,13,14]. (C) Probes used in the RdE-RP, N1E-RP, and N2E-RP 4-plex assays detect 2 SARS-CoV-2 targets and 2 controls. (D) Using our 4-plex assays, E, N1, and N2 give Cq values (mean ± SD, n = 2 experiments) comparable to those for ORF1ab, S, and N from the TaqPath assay (mean of technical triplicates, n = 1 experiment) when detecting cultured SARS-CoV-2. Also, see S1 Data. 2019-nCoV, Novel Coronavirus 2019; Cq, cycle quantification; E, envelope; N, nucleocapsid; ORF, open reading frame; qRT-PCR, quantitative reverse transcription PCR; RdRp, RNA-dependent RNA polymerase; SARS-CoV, Severe Acute Respiratory Syndrome Coronavirus; SARS-CoV-2, Severe Acute Respiratory Syndrome Coronavirus 2; SD, standard deviation.

samples with no or little usable patient material is one possible explanation for this [15]. In essence, the *RPP30* control provides a measure of sample quality. In addition, a defined amount of PhHV is spiked into each sample with the lysis buffer at the start of the nucleic acid isolation procedure, resulting in a known cycle quantification or Cq value (also referred to as the cycle threshold or Ct value, [16]). Detection of PhHV (using the glycoprotein B gene as a target) simultaneously controls for extraction and amplification efficiency and indicates absence of PCR inhibitors [17,18].

If paired with an in-house RNA extraction protocol, our assay can be performed for less than £2 GBP ($2.50 USD) per test, excluding cost of plastic consumables, which could mean a potential 10-fold difference in cost compared to commercial kits. Here, we present data that demonstrate equivalent performance to the commercial TaqPath COVID-19 CE-IVD RT-PCR Kit (Thermo Fisher Scientific, Pleasanton, California, USA) and Abbott RealTime SARS-CoV-2 assay (Abbott Molecular, Des Plaines, Illinois, USA). We also document the utility of inclusion of RPP30 as a human internal control to provide important sample quality information.

## Results

### N and E gene multiplex assays sensitively detect viral RNA

With the aim of developing a sensitive and affordable assay for the detection of SARS-CoV-2 RNA, we tested 3 different 4-plex strategies (Fig 1B and 1C). All made use of RPP30 (HEX) and PhHV (Cy5) probes employed as internal controls (controlling for the presence of human cells in patient samples and successful nucleic acid isolation, respectively), as well as a CFR 610-labelled probe for the SARS-CoV-2 E gene. To enhance assay sensitivity and specificity, a FAM-labelled probe against a second SARS-CoV-2 target was included in each of the 3 assays: RdRp, N1, or N2 (N gene). We named these tests RdE-RP (RdRp, E, RPP30, PhHV), N1E-RP (N1, E, RPP30, PhHV), and N2E-RP (N2, E, RPP30, PhHV), respectively (Fig 1C). Initial validation tests showed that these assays were capable of detecting cultured SARS-CoV-2, with Cq values for E, N1, and N2 similar to those for ORF1ab, S, and N gene obtained using the Thermo Fisher Scientific TaqPath COVID-19 assay. In contrast, RdRp Cq values were substantially higher (Fig 1D; 4.4 to 6.1 cycles above the other targets).

Next, the RdE-RP, N1E-RP, and N2E-RP assays were used to test SARS-CoV-2–positive and SARS-CoV-2–negative patient samples ($n$ = 19), comparing them to the commercial TaqPath assay and Abbott RealTime SARS-CoV-2 assay (which detects RdRp and N gene). The N1E-RP and N2E-RP assays both correctly identified all 9 samples that had tested positive using the TaqPath and Abbott assays (Table 1). The RdE-RP assay performed less well, identifying 7 of these samples correctly, giving inconclusive results for the other 2 (P18 and 19), with E gene but not RdRp detected. Overall, we find RdRp detection to be at least 20-fold less sensitive than for E gene, N1, and N2 under our assay conditions, consistent with reports by others [19]. This may be due to a mismatch in the reverse primer employed in the RdRp (P2) assay, as originally designed [14]. Both N1E-RP and N2E-RP assays also identified positive samples that scored negative with the commercial tests, suggesting potentially higher sensitivity of our assays. Of the 10 patient samples that were negative for the Abbott assay, 9 were similarly shown to be negative using the N1E-RP assay, whereas 8 of these were negative for the N2E-RP assay. Patient 11, previously negative using the Abbott assay, was inconclusive with TaqPath (1 of 3 SARS-CoV-2 targets detected) and N2E-RP assays (1 of 2 targets detected), but positive in the N1E-RP assay. Patient 12 had previously tested negative using both Abbott and TaqPath assays and was also negative for N1E-RP; however, this sample tested weakly positive for both SARS-CoV-2 targets in the N2E-RP assay. Cq values were high for both P11 and P12, close to the limit of detection, but with multiple viral targets detected these likely represent true positives. However, differentiating between samples with low viral loads and false positives is challenging. Analysis of such samples by Sanger sequencing of PCR products or nanopore sequencing of RNA present could provide useful information. Further clinical evaluation and repeat sampling of the patient involved may also be a beneficial route to a secure clinical diagnosis.

**Table 1. The multiplex assay detecting RdRp and E gene (RdE-RP) is not sufficiently sensitive, whereas assays detecting N and E gene (N1E-RP and N2E-RP) accurately identify positive patient samples.**

| | RdE-RP | | N1E-RP | | N2E-RP | | All assays | TaqPath assay | | | Abbott | Conclusion | | | | |
|---|---|---|---|---|---|---|---|---|---|---|---|---|---|---|---|---|
| Patient | RdRp | E | N1 | E | N2 | E | RPP30 | N | ORF1ab | S | Cq [a] | RdE-RP | N1E-RP | N2E-RP | Taq Path | Abbott |
| P1 | UD | UD | UD | UD | UD | UD | 28.2 ± 0.56 | UD | UD | UD | UD | N | N | N | N | N |
| P2 | UD | UD | UD | UD | UD | UD | 31.8 ± 0.58 | UD | UD | UD | UD | N | N | N | N | N |
| P3 | UD | UD | UD | UD | UD | UD | 29.3 ± 0.54 | UD | UD | UD | UD | N | N | N | N | N |
| P4 | UD | UD | UD | UD | UD | UD | 27.8 ± 0.51 | UD | UD | UD | UD | N | N | N | N | N |
| P5 | 24.17 | 18.72 | 19.33 | 18.19 | 18.16 | 17.50 | 22.7 ± 1.37 | 20.09 | 20.03 | 20.73 | 18.26 | P | P | P | P | P |
| P6 | 25.39 | 20.58 | 21.55 | 19.90 | 20.28 | 20.27 | 24.9 ± 1.65 | 22.80 | 22.04 | 22.78 | 21.39 | P | P | P | P | P |
| P7 | 28.65 | 24.23 | 25.09 | 23.41 | 23.97 | 22.85 | 27.1 ± 0.79 | 25.94 | 25.64 | 26.27 | 23.70 | P | P | P | P | P |
| P8 | 32.33 | 25.74 | 26.09 | 25.26 | 24.92 | 24.56 | 27.6 ± 0.49 | 27.00 | 27.19 | 27.90 | 25.78 | P | P | P | P | P |
| P9 | UD | UD | UD | UD | UD | UD | 28.2 ± 0.29 | UD | UD | UD | UD | N | N | N | N | N |
| P10 | UD | UD | UD | UD | UD | UD | 27.9 ± 0.48 | UD | UD | UD | UD | N | N | N | N | N |
| P11 | UD | UD | 37.98 | 35.33 | UD | 33.95 | 30.6 ± 0.49 | UD | 39.42 | UD | UD | N | P | Inc | Inc | N |
| P12 | UD | UD | UD | UD | 38.03 | 34.81 | 27.5 ± 0.33 | UD | UD | UD | UD | N | N | P | N | N |
| P13 | 24.85 | 20.26 | 20.76 | 19.54 | 19.73 | 18.94 | 24.1 ± 1.64 | 21.35 | 21.85 | 22.41 | 19.84 | P | P | P | P | P |
| P14 | UD | UD | UD | UD | UD | UD | 27.9 ± 0.49 | UD | UD | UD | UD | N | N | N | N | N |
| P15 | 31.64 | 25.50 | 27.06 | 24.94 | 25.43 | 24.37 | 25.9 ± 0.38 | 27.50 | 26.73 | 27.30 | 25.27 | P | P | P | P | P |
| P16 | UD | UD | UD | UD | UD | UD | 27.9 ± 0.36 | UD | UD | UD | UD | N | N | N | N | N |
| P17 | 26.97 | 17.44 | 16.08 | 17.23 | 16.93 | 16.85 | 21.9 ± 1.67 | 16.06 | 17.02 | 18.46 | 14.16 | P | P | P | P | P |
| P18 | UD | 29.77 | 31.10 | 29.17 | 30.28 | 29.20 | 29.7 ± 0.43 | 31.54 | 30.71 | 31.30 | 30.32 | Inc | P | P | P | P |
| P19 | UD | 31.76 | 34.03 | 31.13 | 32.40 | 30.21 | 27.4 ± 0.63 | 33.87 | 33.71 | 35.22 | 32.49 | Inc | P | P | P | P |

Values used for Figs 3 and 4 and S2B and S4 Figs.

[a]The output for the Abbott test, which detects RdRp and N gene, is given in a single CN value, which is approximately equivalent to Cq minus 10 (so Cq = CN + 10).

CN, cycle number; Cq, cycle quantification; E, envelope; Inc, inconclusive; N, negative; P, positive; RdRp, RNA-dependent RNA polymerase; UD, undetermined.

As our initial characterisation demonstrated the N1E-RP and N2E-RP assays to be at least as sensitive as 2 commercial assays, TaqPath COVID-19 CE-IVD RT-PCR Kit (Thermo Fisher Scientific) and Abbott RealTime SARS-CoV-2 (Abbott Laboratories), we focused on these 2 assays for further validation experiments.

## Multiplex assays for N and E genes can detect between 1 and 50 copies of IVT RNA

To determine the detection limit for the N1E-RP and N2E-RP assays, in vitro transcribed (IVT) RNA controls for each of the SARS-CoV-2 targets were prepared. An equimolar mix was used to make a dilution series (for 10,000 down to 1 copy of RNA per reaction; a 50-copy control was also included) and Cq values determined in triplicate using both assays (Fig 2A). To test our nucleic acid isolation protocol (S2 Protocol), all dilutions after extraction were simultaneously tested in triplicate (S1A Fig). All probes (E, N1, and N2) reproducibly detected 50 copies (Fig 2A, S1A Fig, S1 Table). The N1 probe detected 10 copies reproducibly (6 out of 6; 6/6), while the N2 probe did so in some reactions (4/6) of the respective 4-plex assays. The E probe detected 10 copies reproducibly in the N2E-RP assay (6/6), but only did so in half of the N1E-RP reactions (3/6). As might be expected, single copies of RNA were only detected in a small proportion of reactions for each probe: E (2/12), N1 (3/6), and N2 (1/6). We therefore conclude that our assays have the sensitivity to detect between 1 and 50 copies of IVT RNA (Fig 2A, S1A Fig, S1 Table), both pre and post-extraction.

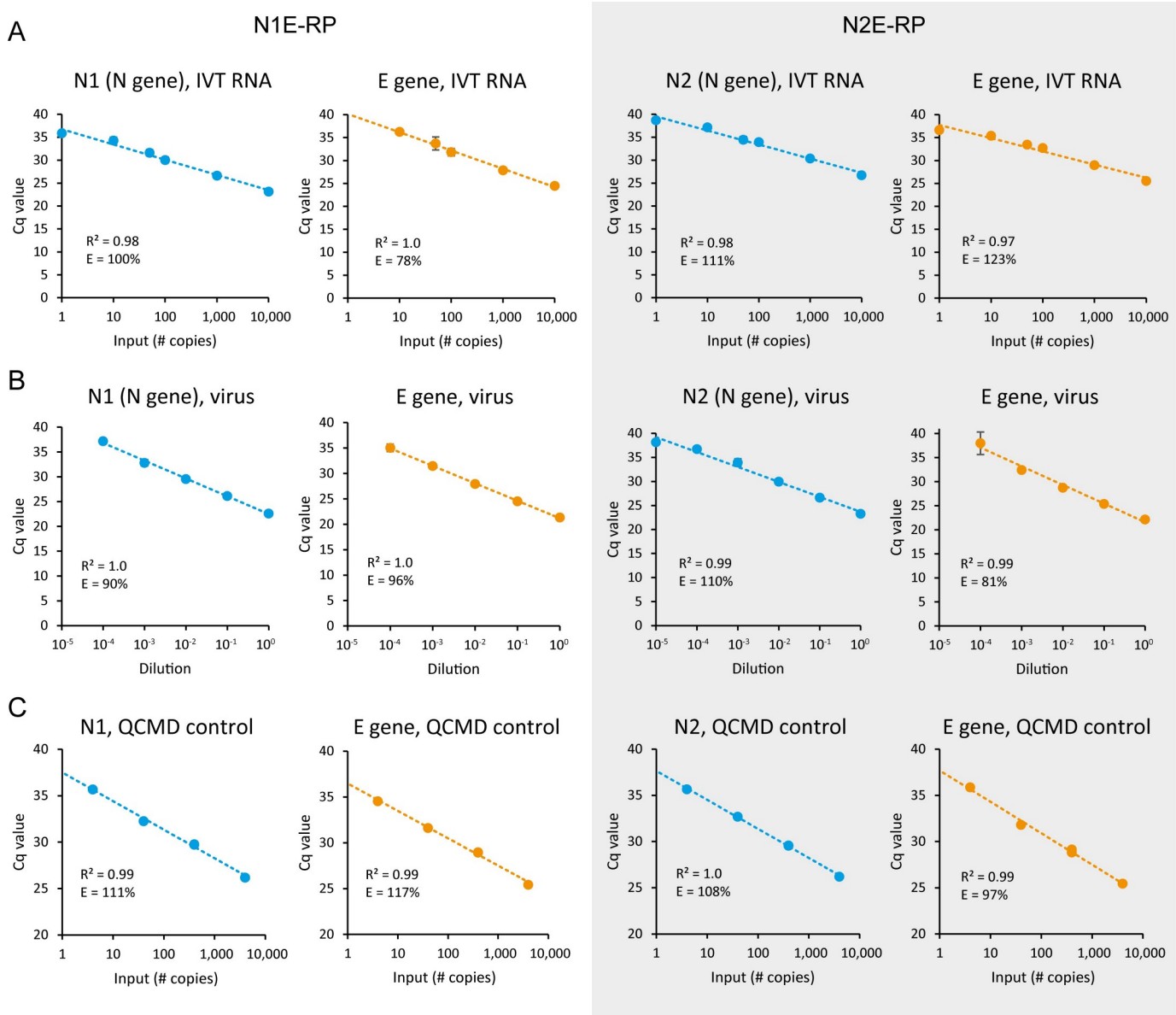

**Fig 2. The N1E-RP and N2E-RP 4-plex assays detect between 1 and 50 copies of SARS-CoV-2 RNA.** N1E-RP and N2E-RP RT-qPCR assays were performed with the following controls: (A) 1 to 10,000 copies of SARS-CoV-2 control RNA (IVT); (B) Serial dilution of RNA isolated from cultured SARS-CoV-2 (hCoV-19/England/02/2020). Mean ± SD for technical triplicates for (A) and (B). (C) RNA isolated from QCMD viral controls (BetaCoV/Munich/ChVir984/2020). These controls contained different amounts of SARS-CoV-2 virus at known copy number (see Materials and methods for details; also see S3 Table). $R^2$ values for logarithmic trend line fitting; E, amplification efficiency (see Materials and methods). Also, see S1–S3 Tables and S1 Data. BetaCoV, Betacoronavirus; E, envelope; hCoV-19, Human Coronavirus 2019; IVT, in vitro transcribed; N, nucleocapsid; QCMD, Quality Control for Molecular Diagnostics; RT-qPCR, quantitative reverse transcription PCR; SARS-CoV-2, Severe Acute Respiratory Syndrome Coronavirus 2; SD, standard deviation.

## N1E-RP and N2E-RP assays can detect between 1 and 3 copies of viral RNA

To confirm the range of detection for total viral RNA, nucleic acids isolated from cultured SARS-CoV-2 were used to make a dilution series ($10^{-1}$ to $10^{-6}$) and Cq values determined in triplicate using the N1E-RP and N2E-RP assays (Fig 2B). To test the impact of the nucleic acid isolation procedure on extraction of low copy numbers of viral RNA and to test the PhHV

spike-in control, the dilution series was also re-extracted and used for RT-qPCR simulta-neously (S1B Fig). Sensitivity of detection for these samples was highest for E gene, followed by N1 and N2 (Fig 2B, S1B Fig, S2 Table). Signal was lost for the $10^{-5}$ dilution in most cases, consistent with the Cq values of the undiluted sample (21.3 to 23.4) and the 100,000-fold reduction in copy number for this dilution (theoretically predicted Cq values, approximately 38 to 40). For all extractions and RT-qPCR replicates, the signal for the PhHV spike-in was highly reproducible (S2 Table), with a Cq value of 32.5 ± 0.40 (mean ± SD, range 30.7 to 33.0), indicating robust extraction efficiency and absence of PCR inhibitors.

Finally, 8 quality control samples obtained from Quality Control for Molecular Diagnostics (QCMD; an international external quality assessment organisation) were also tested using the N1E-RP, N2E-RP, and TaqPath assays. Each assay gave the same outcome, consistent with data provided by QCMD (S3 Table), identifying 5 samples as positive (4 to 4,000 copies of SARS-CoV-2 per reaction) and 3 as negative (containing transport medium or different coro-naviruses). All probes used in the N1E-RP and N2E-RP assays displayed a linear range of detection down to 4 copies of viral RNA (Fig 2C). These calibration curves were retrospectively used to calculate the detection limit of our own viral RNA serial dilution (Fig 2B), showing that in these experiments, our assays detected between 1 and 3 copies of SARS-CoV-2 genomic RNA.

## N1E-RP and N2E-RP multiplex assays correctly identify positive patient samples

Next, to further establish assay reproducibility in a diagnostics context, the N1E-RP and N2E-RP assays were performed on an additional 89 patient samples and results compared to the TaqPath assay. The patient samples contained both SARS-CoV-2–positives and SARS-CoV-2–negatives and were tested blind. Internal controls were included to provide confirmation of successful nucleic acid extraction and absence of PCR inhibitors, with lysis buffer spiked with both MS2 (an RNA bacteriophage that infects *Escherichia coli*) and PhHV (a DNA virus that infects seals), detected by the TaqPath and N1E-RP/N2E-RP assays, respec-tively. In addition, the same 3 controls were performed for each assay: an extracted viral trans-port medium control (negative for SARS-CoV-2 and *RPP30* and positive for PhHV), a non-extracted water only control (negative for all targets), and a non-extracted IVT RNA positive control (50 copies; positive for SARS-CoV-2 and negative for *RPP30* and PhHV).

Results for controls were as anticipated (S4 Table), with signal absent (undetermined) for SARS-CoV-2 and *RPP30* targets for the negative controls and Cq values for the SARS-CoV-2 RNA positive control (50 copies) similar to those obtained previously (Fig 2A). The PhHV control gave consistent Cq values for both N1E-RP (32.5 ± 1.1) and N2E-RP assays (33.3 ± 1.2; S2A Fig, S4 Table), confirming reliable and reproducible extraction of nucleic acids from patient samples; similar to the MS2 control used in the TaqPath assay (mean Cq value, 25.6 ± 0.9; S2A Fig, S4 Table). Out of the 89 samples, the TaqPath assay identified 75 samples as negative, 1 as inconclusive, and 13 as positive. Both the N1E-RP and N2E-RP assay detected the same 13 positive samples, and the majority of TaqPath negative samples were similarly negative in our assays (*n* = 74). For the N1E-RP assay, 6 of the negative samples had Cq values between 39.0 and 43.2 for N1 (E gene not detected), suggesting potentially higher sensitivity of the N1 probe in this assay. The sample that was inconclusive with the TaqPath assay (P75) was positive for both N1E-RP and N2E-RP assays, consistent with this being a true positive. In addition, there was 1 sample (P53) that was negative with TaqPath, but positive for both N1E-RP and N2E-RP assays, albeit with very high Cq values (between 35.7 and 39.2), close to the limit of detection.

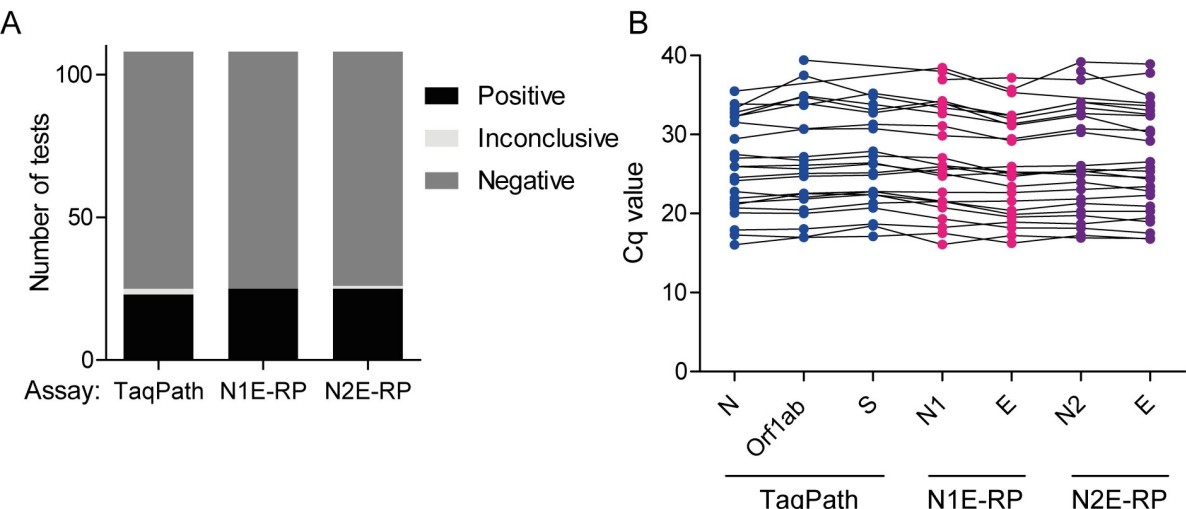

**Fig 3. The N1E-RP and N2E-RP 4-plex assays perform similarly to the TaqPath assay, correctly identifying positive and negative patient samples.** (A) The Taqpath, N1E-RP, and N2E-RP assays each identified a similar number of positives and negatives among 108 patient samples. Inconclusive: Only 1 of the SARS-CoV-2 targets was detected. (B) Cq values for each of the SARS-CoV-2 targets in the TaqPath (N, Orf1ab, and S), N1E-RP (N1 and E), and N2E-RP (N2 and E) assays were comparable (for n = 24–26 positive patients). Also, see Table 1, S4 Table, and S1 Data. Cq, cycle quantification; E, envelope; N, nucleocapsid; SARS-CoV-2, Severe Acute Respiratory Syndrome Coronavirus 2.

Altogether, our data (for $n$ = 108 patient samples) establish that the sensitivity of the N1E-RP and N2E-RP assays is similar to, if not higher than, the TaqPath assay (Fig 3).

## Substantial variability in NTS quality, as measured by human *RPP30*, impacts on assay sensitivity

The range of Cq values for the human *RPP30* control was much greater than that of the PhHV internal control (Fig 4A and 4B, S2 Fig, S4 Table). This indicated that there was considerable variability in the amount of cellular material present in different patient samples. The *RPP30* primer/probe set has good amplification efficiency and was able to detect 10 copies of positive control nucleic acids (S3 Fig), hence Cq values for this probe represent a good measure of the presence of intact cellular nucleic acids in patient samples. Although *RPP30* was detected in all samples, Cq values ranged from 20.1 to 32.1 for the N1E-RP assay and from 20.3 to 32.4 for the N2E-RP assay, which equates to an approximately 4,000-fold difference in extracted nucleic acids between the best and worst samples. Although the distribution of positives (Fig 4A and 4B) shows that samples with high *RPP30* Cq values can still test positive, this is not an adequate measure of the likelihood of false negatives among samples with similarly low levels of human material. A statistically significant linear correlation between Cq values for each of the viral probes (E, N1, and N2) and the Cq values for the *RPP30* sample quality probe ($p < 0.001$; Fig 4C, S1 Data) established that samples containing fewer human cells are more likely to have less SARS-CoV-2, potentially decreasing the chance of detection. To visually demonstrate the impact of this, we normalised SARS-CoV-2 Cq values to the sample with the lowest amount of human material detected (reference sample, *RPP30* Cq value 32.1 and 32.4 for N1E-RP and N2E-RP assays, respectively). Due to the linear correlation between *RPP30* and viral Cq values, subtracting the difference in *RPP30* Cq between a particular positive sample and the reference sample from the SARS-CoV-2 Cq gives an indication of what the viral Cq may have been had it contained the same low amount of human material present in the reference sample. This shows that of the 26 positive samples we detected, between 4 and 6 (15%

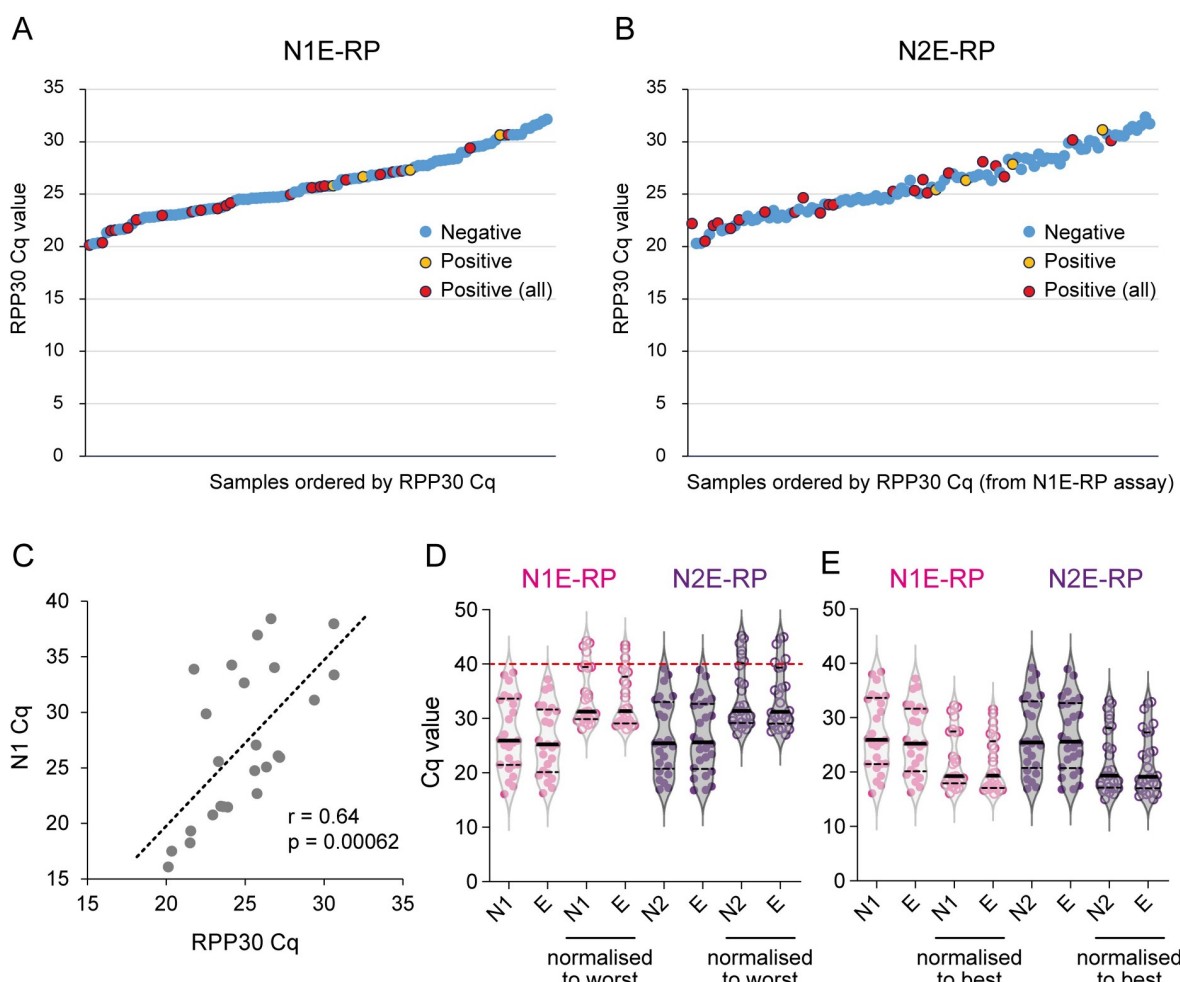

**Fig 4. The *RPP30* control indicates substantial variability in sample quantity/quality, impacting on assay sensitivity.** (A, B) *RPP30* Cq values for 108 patient samples ranked from low to high (based on N1E-RP ranks) for N1E-RP (A) and N2E-RP (B) assays. SARS-CoV-2–positives (red, positive in N1E-RP, N2E-RP, and TaqPath assays; orange, positive in N1E-RP and/or N2E-RP assays only) are detected for samples with low as well as high *RPP30* Cq values. (C) *RPP30* and SARS-CoV-2 Cq values are highly correlated, demonstrating that samples with fewer human cells have lower levels of SARS-CoV-2, supporting the validity of *RPP30* Cq as a measure of sample quality. r, Pearson correlation coefficient; *p*-value calculated by F-test. (D, E) Normalisation to *RPP30* levels increases clustering of viral Cq values. Reduced variability in apparent SARS-CoV-2 levels when normalising to the worst (highest *RPP30* Cq; D) as well as the best (lowest *RPP30* Cq; E) sample. Plots in panel (D) demonstrate the impact of sample quality on assay sensitivity, with 4–6 positives (15%–23%) below the detection limit (above the red line, viral Cq > 40) for a worst quality sample scenario. Also, see Table 1, S4 Table, and S1 Data. Cq, cycle quantification; E, envelope; SARS-CoV-2, Severe Acute Respiratory Syndrome Coronavirus 2.

to 23%) would not have tested positive, with at least 1 of the viral targets exceeding the detection limit (Cq > 40; Fig 4D, S4A Fig). Theoretically, using this approach, even a strong positive sample (SARS-CoV-2 Cq value of 28) of good quality (*RPP30* Cq value of 20.1) may have given a false-negative test result (SARS-CoV-2 Cq value of 40) if it had contained the same low amount of human material as the reference sample (*RPP30* Cq value of 32.1; viral Cq: 32.1 − 20.1 + 28 = 40). Conversely, normalising samples to an optimal quality sample (RPP30 Cq 20.1/20.3 for N1E-RP/N2E-RP) gives an indication of what viral Cq values may have been if all samples had contained a similar (more optimal) amount of material (Fig 4E, S4B Fig). This highlights the possibility that a proportion of apparent SARS-CoV-2–negative samples are in fact false negatives as a result of insufficient material in the swab fluid. Notably, the SARS-CoV-2 Cq values clustered more strongly after normalisation (Fig 4D and 4E, S4 Fig). This

reduced variability not only shows that the amount of human material present in NTS samples impacts on assay sensitivity, but also suggests that variability in viral load is not as great as implied by RT-qPCR data without normalisation.

## Discussion

Here, we describe a user-friendly protocol (S1 and S2 Protocols) for an accurate and affordable SARS-CoV-2 RT-qPCR test. Although we did not detect substantial differences between our 2 assays, others have reported higher sensitivity of the N1 over the N2 assay [19]. We therefore recommend the use of the N1E-RP assay for primary testing, whilst the N2E-RP assay could be employed if initial results are inconclusive. We provide detailed materials and methods to enable others to rapidly set up this assay in their own laboratory or to adapt it to locally avail-able equipment and reagents. While we provide extensive validation of the reagents and instru-ments used to perform these multiplex RT-qPCR assays, our methods allow some flexibility. Probes with different labels as well as alternative real-time PCR machines could be used, as long as the different dyes can be detected simultaneously. Also, spike-in controls other than PhHV could be used, such as recombinant DNA or RNA. However, a virus control would bet-ter mimic extraction of SARS-CoV-2 RNA, and other viruses with an RNA genome (e.g., lenti-virus, routinely produced in many molecular biology laboratories) would make particularly good controls, not only confirming successful RNA extraction, but also controlling for RNA stability and reverse transcription. As a further development, replacing E gene with M or S gene probes could provide N2M or N2S assays as fully independent second-line tests. Both M and S gene assays have been shown to have high sensitivity [20,21], and their target regions dis-play low sequence variation (S5 Table). Additional improvements to our protocol could include the use of control primers/probe specific to a human RNA transcript (the *RPP30* prim-ers/probe described here detect both RNA and genomic DNA) as this would ensure that sam-ples contain intact RNA [22]. However, it should be stressed that any changes to the protocol may also change the sensitivity of SARS-CoV-2 detection, and new protocols should undergo appropriate validation before use for diagnostic purposes.

Our assays have high analytical sensitivity, equivalent to commercial CE-IVD kits. RT-qPCR tests are molecular tests with high intrinsic accuracy; however, false-positive and false-negative results can occur. The use of multiplex assays that detect multiple SARS-CoV-2 tar-gets, such as those reported here, reduces the chance of both. Off-target reactivity is one possi-ble cause of false positives, and although some have reported high false-positive rates for the E gene assay [20,23], this does not match our experience, with high concordance between N1, N2, and E gene results in our patient cohort. In 2 patients, our N1E-RP and N2E-RP assays detected virus, albeit weakly, whereas commercial assays did not. As multiple SARS-CoV-2 targets were positive, these are likely true positive results and not due to off-target reactivity. False positives can also occur due to lab issues such as sample mislabelling, data entry errors, reagent contamination with target nucleic acids, or contamination of primary specimens. However, high standards of quality control at all stages of testing and effective mitigation strat-egies should quickly identify problems. Additionally, sample retesting with an independent assay and/or patient resampling should also be effective measures to counter false positives, particularly in low pretest probability situations such as mass screening.

False-negative test results are an important ongoing issue, estimated to be somewhere between 2% and 54% [24–26]. Sequence variation at primer/probe binding sites could be one factor resulting in false negatives. However, for the primers and probes used, the chance of this is low (>97.6% of 97,782 strains have no relevant changes; Table 2, S5 Table), and strains with mutations in 2 independent targets should be very rare. Therefore, sequence variation is not

**Table 2. Percentage of known SARS-CoV-2 genomic sequences with mutations in primer/probe binding sites for E gene, N1, and N2 assays.**

| Assay | Primer/probe | Percentage of strains with any mismatch/deletion[a] | | Percentage of strains with mismatch in 5 most 3′ nt[a] | |
|---|---|---|---|---|---|
| | | Per primer/probe | Per assay[b] | Per primer | Per assay[c] |
| E gene | E_Sarbeco-F1 | 0.10% | 0.26% | 0.006% | 0.008% |
| | E_Sarbeco-P1 | 0.12% | | (0.009%) | |
| | E_Sarbeco-R2 | 0.04% | | 0.002% | |
| N1 | 2019-nCoV_N1-F | 0.29% | 2.39% | 0.034% | 0.157% |
| | 2019-nCoV_N1-P | 1.79% | | (0.057%) | |
| | 2019-nCoV_N1-R | 0.33% | | 0.124% | |
| N2 | 2019-nCoV_N2-F | 0.35% | 1.82% | 0.153% | 0.169% |
| | 2019-nCoV_N2-P | 0.33% | | (0.008%) | |
| | 2019-nCoV_N2-R | 1.19% | | 0.015% | |

High-quality genome sequences analysed only ($n$ = 97,782); also, see S1 Data.

[a]A total of 0%–0.003% of genomes have deletions in primer/probe binding regions, so majority are mismatches. Presence of these changes does not necessarily impact on primer/probe performance. Mutations at the 3′ ends for the primer regions (here defined as the final 5 nt) are more likely to affect assay sensitivity.

[b]Mismatches/deletions in the primer/probe set used in single-target assay.

[c]Mismatches in the forward and reverse primers only. For probes, values are given between brackets for reference (in the "per primer" column).

2019-nCoV, Novel Coronavirus 2019; E, envelope; N, nucleocapsid; nt, nucleotide; SARS-CoV-2, Severe Acute Respiratory Syndrome Coronavirus 2.

expected to be a significant contributing factor to the number of false negatives. In contrast, low sample quality provides a much more likely explanation, and this may be particularly important in case of self-sampling. Systematic inclusion of a human cellular control to provide sample quality metrics could therefore have utility in reducing the number of false negatives. Testing saliva, as an alternative to NTS sampling, could also be beneficial as a modality that may have less sample to sample variability [7].

Absence of *RPP30* signal (undetected or Cq >40) clearly indicates that absence of viral detection cannot be interpreted as a negative test result and that a repeat test is required (S6 Table). However, utilising *RPP30* Cq values when interpreting an apparent SARS-CoV-2–negative sample requires further consideration: What should the *RPP30* Cq limit be for which to order a repeat test? One option would be to simply set an arbitrary cutoff, e.g., one could decide to retest any samples with *RPP30* Cq >30 or with Cq values above the 95th centile (Cq approximately 31 for our 108 samples). To determine robust cutoff limits, collection of *RPP30* data for a much larger number of patient samples would be desirable. This would allow development of diagnostic algorithms that could incorporate a sample quality score based on the level of *RPP30* detected. Nonetheless, *RPP30* data, even as it stands, are useful for the interpretation of cases for which only 1 of the SARS-CoV-2 targets is (weakly) positive, with samples with high *RPP30* Cq values interpreted with particular caution. In such cases, repeat testing of the same sample (with an independent assay of equal or better sensitivity) would be advisable, and repeat patient specimen collection and testing might also be considered (see S6 Table for guidance). In addition, ongoing monitoring of swab quality allows rapid identification of potential technical issues with swabbing. Finally, normalisation of viral Cq values using *RPP30* Cq values could be helpful in a research context to derive a more meaningful measure of viral loads by removing one source of variability, e.g., when monitoring changes in patients over time and/or in response to treatments.

Ultimately, the clinical sensitivity of any diagnostic test is influenced by multiple factors, including sample timing relative to symptom onset, sample type, and sample quality. The inclusion of a human control in our assays provides an internal sample quality control,

supporting improved interpretation of test results, which could contribute to reducing false-negative rates.

Taken together, we show that sensitive and robust RT-qPCR assays for the detection of SARS-CoV-2 are available at a fraction of the cost of comparable commercial assays. The use of these assays could make widespread population testing more feasible.

## Materials and methods

### Patient samples

Samples were collected from symptomatic individuals by trained healthcare professionals using combined nose and throat swabbing and processed for diagnostic testing using validated CE-IVD assays. Excess samples were then used to validate the in-house multiplex assays. A variety of swabs and viral transport media (VTM) were used. In each case, swabs were placed in VTM and kept at ambient temperature until processed (within 24 hours).

### Ethics statement

After diagnostic testing using validated CE-IVD assays, excess samples were used to validate in-house multiplex assays, with specimen anonymisation by coding, compliant with Tissue Governance for the South East Scotland Scottish Academic Health Sciences Collaboration Human Annotated BioResource (reference no. SR1452).

### Nucleic acid isolation

Nucleic acids were isolated using the Omega Mag-Bind Viral DNA/RNA 96 Kit (Omega Biotek, Norcross, Georgia, USA; Cat. No. M6246), following the Supplementary Protocol for NP Swabs (April 2020 version). Briefly, 200 μl VTM was taken from patient swab sample inside a Class 2 Safety Cabinet and mixed with 240 μl TNA Lysis Buffer, 1 μl carrier RNA, and extraction controls (MS2, provided as part of the TaqPath COVID-19 Kit and PhHV, provided by the laboratory of Jürgen Haas) in screw capped tubes, for virus inactivation. After incubation at room temperature for at least 15 minutes, samples were transferred from tubes into 96-well KingFisher Deep well plates (Thermo Fisher Scientific, Waltham, Massachusetts, USA; Cat. No. 95040450) containing 280 μl isopropanol and 2 μl Mag-Bind Particles per well, using a Biomek NX$^P$ Automated Liquid Handler (Beckman Coulter, High Wycombe, United Kingdom). Plates were then moved and the isolation completed on a KingFisher Flex robot (Cat. No. 5400610) as instructed by the manufacturer, including washes with 350 μL VHB Buffer and 2× 350 μL SPR Buffer, and RNA finally eluted in 50 μl of nuclease-free water in KingFisher 96-well microplates (Cat. No. 97002540). An in-house version of a magnetic bead-based isolation could further reduce the cost per test, but requires additional validation. For details of RNA isolation protocols, see S2 Protocol.

### Primers and probes

Primers and probes (Table 3) were synthesised and HPLC purified by LGC BioSearch Technologies (Risskov, Denmark) and dissolved in IDTE (10 mM Tris, 0.1 mM EDTA, pH 8.0) to prepare 100 μM stocks. Pre-prepared primer/probe mixes (FAM-labelled) for N1, N2, and *RPP30* were obtained from Integrated DNA Technologies (IDT, USA; Cat. No. 10006713). Since we developed our assay, N1, N2, and *RPP30* primers and probes also became available from IDT as 100 μM stocks, but can also be purchased from other reputable oligonucleotide synthesis companies. All nucleic acid stocks and dilutions were prepared in Eppendorf DNA LoBind tubes (Eppendorf, Hamburgh, Germany; Cat. No. 10051232).

**Table 3. Primer/probe details for 4-plex assays.**

| Target | Oligonucleotide ID | Sequence (5'-3') | Concentration (nM) | PCR product size | Reference |
|---|---|---|---|---|---|
| SARS-CoV-2 E gene | E_Sarbeco_F1 | ACAGGTACGTTAATAGTTAATAGCGT | 400 | 113 bp | [13,14] |
| | E_Sarbeco_R2 | ATATTGCAGCAGTACGCACACA | 400 | | |
| | TxRd_E_Sarbeco_P1 * | CFR-610-ACACTAGCCATCCTTACTGCGCTTCG-BHQ2 | 200 | | |
| SARS-CoV-2 RdRp | RdRp_SARSr-F2 | GTGARATGGTCATGTGTGGCGG | 600 | 100 bp | [13,14] |
| | RdRp-SARSr-R1 | CARATGTTAAASACACTATTAGCATA | 800 | | |
| | FAM_RdRp_SARSr-P2 | FAM-CAGGTGGAACCTCATCAGGAGATGC-BHQ1 | 200 | | |
| SARS-CoV-2 N gene | 2019-nCoV_N1-F | GACCCCAAAATCAGCGAAAT | 500 | 72 bp | [9] |
| | 2019-nCoV_N1-R | TCTGGTTACTGCCAGTTGAATCTG | 500 | | |
| | 2019-nCoV_N1-P | FAM-ACCCCGCATTACGTTTGGTGGACC-BHQ1 | 125 | | |
| SARS-CoV-2 N gene | 2019-nCoV_N2-F | TTACAAACATTGGCCGCAAA | 500 | 67 bp | [9] |
| | 2019-nCoV_N2-R | GCGCGACATTCCGAAGAA | 500 | | |
| | 2019-nCoV_N2-P | FAM-ACAATTTGCCCCCAGCGCTTCAG-BHQ1 | 125 | | |
| Human *RPP30* | Hs_RPP30-F | AGATTTGGACCTGCGAGCG | 500 | 65 bp | [9] |
| | Hs_RPP30-R | GAGCGGCTGTCTCCACAAGT | 500 | | |
| | HEX-Hs_RPP30-P | HEX–TTCTGACCTGAAGGCTCTGCGCG–BHQ1 | 125 | | |
| PhHV-1 glycoprotein B | PhHV-F | GGGCGAATCACAGATTGAATC | 300 | 89 bp | [18] |
| | PhHV-R | GCGGTTCCAAACGTACCA | 300 | | |
| | Cy5-PhHV-P ** | Quasar-670-TTTTTATGTGTCCGCCACCATCTGGATC-BHQ2 | 100 | | |

*Probe named TxRd for simplicity. CFR-610 has virtually identical properties to TexRed.

**Probe named Cy5 for simplicity. Quasar 670 has virtually identical properties to Cy5.

RdRp_SARSr-F2 and RdRp-SARSr-R1, IUPAC codes: S = G or C; R = A or G.

2019-nCoV, Novel Coronavirus 2019; CFR, CAL Fluor Red; E, envelope; IUPAC, International Union of Pure and Applied Chemistry; N, nucleocapsid; PhHV, Phocine Herpes Virus; RdRp, RNA-dependent RNA polymerase; SARS-CoV-2, Severe Acute Respiratory Syndrome Coronavirus 2.

Primer/probe mixes (50×) were prepared for E gene (20 µM E_Sarbeco_F1, 20 µM E_Sarbeco_R2, 10 µM TxRd_E_Sarbeco_P1), RdRp (30 µM RdRp_SARSr-F2, 40 µM RdRp_SARSr-R2, 10 µM FAM_RdRp_SARSr-P2), *RPP30* (25 µM Hs_RPP30-F, 25 µM Hs_RPP30-R, 6.25 µM HEX-Hs_RPP30-P), and PhHV (15 µM PhHV-F, 15 µM PhHV-R, 5 µM Cy5-PhHV-P). The N1 and N2 primers/probes were purchased premixed (approximately 13.3×) from IDT. These individual primer/probe mixes were then used to prepare a single mix for each of the 4-plex assays: 12.5× for RdE-RP (with equal volumes of each of the relevant mixes) and 7.4× for N1E-RP and N2E-RP (with equal volumes of the E, *RPP30*, and PhHV mixes, combined with 3.7× volumes of N1 or N2 mix). Mixes were stored at −20˚C, with working stocks kept at 4˚C.

Primers and probes included in the TaqPath COVID-19 CE-IVD RT-PCR Kit (Thermo Fisher Scientific, Cat. No. A48067) detect SARS-CoV-2 ORF1ab, N, and S gene; those in the Abbott RealTime SARS-CoV-2 assay (Cat. No. 09N77-090) detect RdRp and N gene. Further details are not available, as this information is proprietary.

## RT-qPCR

All RT-qPCRs were performed on Applied Biosystems 7500 Fast Real-Time PCR Systems with ABI 7500 software v2.3 (Thermo Fisher Scientific), using MicroAmp Fast Optical 0.1 mL 96-well reaction plates (Thermo Fisher Scientific, Cat. No. 4346906) and Optical Adhesive film (Thermo Fisher Scientific, Cat. No. 4311971). For our assays, we used the Takara One Step PrimeScript III RT-qPCR kit (Cat. No. RR600B). These were compared to the TaqPath COVID-

19 CE-IVD Kit (Thermo Fisher Scientific, Cat. No. A48067) and the Abbott RealTime SARS--CoV-2 assay (Cat. No. 09N77-090), used as instructed by the manufacturer. The TaqPath assay was performed on the ABI 7500 Fast System, and the Abbott assay was performed on the M2000 system. Experiments using the 4-plex assay were performed as described below, with a user-friendly protocol provided in S1 Protocol.

Reaction master mixes were prepared (20 μl per reaction) for each assay, before adding 5 μl of template RNA per reaction, brief centrifugation, and starting the PCR program. For the RdE-RP 4-plex assay, per reaction 12.5 μl of One-Step mix, 5.5 μl of nuclease-free water, 2 μl of 12.5× primer/probe mix, and 5 μl of template RNA were mixed. For the N1E-RP and N2E-RP 4-plex assays, per reaction 12.5 μl of One-Step mix, 4.16 μl of nuclease-free water, 3.34 μl of 7.4× primer/probe mix, and 5 μl of template RNA were mixed. For all 4-plex reactions, the PCR program was 5 minutes at 52°C, 10 seconds at 95°C, then 45 cycles of 3 seconds at 95°C and 30 seconds at 60°C. For detection, the FAM, JOE, TEXAS RED and CY5 channels were used.

## Amplification efficiency

Amplification efficiency (E) was determined for all standard curves using the slope of the linear curve fit when plotting Cq values versus the log of input amounts. An E of 100% means that the number of molecules of the target sequence double during each replication cycle. The following equation was used:

$$E = 100 \times (10^{-1/\text{slope}} - 1) \tag{1}$$

## Positive controls

Positive control RNAs generated by in vitro transcription were provided by Sylvie Behillil (Institut Pasteur, Paris, France) for E gene [9] and by Christine Tait-Burkard (Roslin Institute, Edinburgh, UK) for RdRp [13,14], N1, and N2 [9]. An equimolar mix of all RNAs was prepared at $2.5 \times 10^8$ copies/μl and aliquots stored at −80°C. Dilution series were prepared in nuclease-free water, in Eppendorf DNA LoBind tubes (Cat. No. 10051232), at 2,000, 200, 20, 10, 2, and 0.2 copies/μl.

A cultured SARS-CoV-2 control (strain hCoV-19/England/02/2020; GISAID Accession EPI_ISL_407073, [27–29]) was provided by Rory Gunson (NHS Molecular Development in Virology and Microbiology, Glasgow, UK). Except for the reported S15519T mismatch in RdRp-SARSr_R1 [19], this strain has no mutations in target sites for the primers and probes used in our assays [30].

## QCMD controls

QCMD (Glasgow, UK) provided controls as part of the "QCMD 2020 Coronavirus Outbreak Preparedness (CVOP) EQA Pilot Scheme" [31]. RNA extractions were performed using 200 μl of each sample, eluting in 50 μl. After samples were tested blind with our assays, expected results along with sample identities were provided by QCMD. Quantification of control samples was carried out by QCMD prior to distribution within the EQA scheme, using droplet digital PCR (ddPCR) with E-gene primers and probe [13,14] on the Bio-Rad ddPCR platform (Bio-Rad Laboratories, Hercules, California, USA). A serial dilution of inactivated SARS-CoV-2 (strain BetaCoV/Munich/ChVir984/2020; GenBank Accession MT270112, [32]) was prepared, and each dilution replicate was tested 4 times using both RT-qPCR and ddPCR assays. Regression analysis was used to assess the linearity across the dilution series, and the analytical measurement range established for both assays, comparing results of each by Bland–Altman

difference plot. Except for the reported S15519T mismatch in RdRp-SARSr_R1 [19], this strain has no mutations in target sites for the primers and probes used in our assays [30]. Concentrations determined by ddPCR in Log10 dPCR copies/ml were used to calculate the number of copies of input RNA.

## Statistical analysis

To determine whether a linear relationship exists between the observed Cq values for viral probes and the *RPP30* (sample quality) probe, we used Pearson correlation coefficient and linear regression. Linear regression *p*-values were calculated using an F-test. Qualitative visual model diagnostics indicated in each case that the statistical assumptions of linear regression models were not violated, in particular the normality and homoscedasticity of residuals. Statistical analysis was performed using R version 4.0.3 [33].

## Primer/Probe mismatch analysis

SARS-CoV-2 (hCoV-19) genome sequences and multiple sequence alignment (MSA) of 131,759 strains were downloaded from the GISAID EpiCov™ database [27,29]. Local alignments were generated between each oligo and the hCoV-19/Wuhan/WIV04/2019 reference strain using biopython's pairwise2 module [34]. Alignment coordinates were then transposed to the corresponding positions in the MSA. For each strain sequence in the MSA, mismatches and gaps were counted, where mixed bases (International Union of Pure and Applied Chemistry [IUPAC] ambiguity codes with the exception of N) were present in either oligo or strain sequence a position was considered to "match" if there was overlap between the mixed bases. If either oligo sequence or strain sequence contained gaps relative to one another, a pairwise local alignment was performed between the oligo and the strain sequence corresponding to the oligo position ± 20 flanking nucleotides in order to detect any ungapped matches between strain sequence and oligo.

To ensure only high-quality sequences were included in the analysis, genome sequences with >1% Ns or with gaps ≥ 95th percentile were excluded. Sequences with Ns in oligo regions were also excluded, leaving a total of 97,782 sequences. Code used to count mismatches between strains and oligos is provided at https://github.com/david-a-parry/SARS-CoV-2_oligos_vs_strains.

## Supporting information

**S1 Table. N1E-RP and N2E-RP assay Cq values for SARS-CoV-2 RNA controls (1 to 10,000 copies) pre- and post-extraction.** Values used for Fig 2A and S1A Fig. Cq, cycle quantification; SARS-CoV-2, Severe Acute Respiratory Syndrome Coronavirus 2.
(PDF)

**S2 Table. N1E-RP and N2E-RP assay Cq values for cultured SARS-CoV-2 dilution series (before and after re-extraction).** Values used for Fig 2B and S1B Fig. Cq, cycle quantification; SARS-CoV-2, Severe Acute Respiratory Syndrome Coronavirus 2.
(PDF)

**S3 Table. N1E-RP, N2E-RP, and TaqPath assays all correctly identify SARS-CoV-2–positive QCMD quality control samples.** Values used for Fig 2C. Cq, cycle quantification; QCMD, Quality Control for Molecular Diagnostics; SARS-CoV-2, Severe Acute Respiratory Syndrome Coronavirus 2.
(PDF)

**S4 Table. The N1E-RP and N2E-RP 4-plex assays perform as well as the TaqPath CE-IVD assay on patient samples.** Values used for Figs 3 and 4 and S2 and S4 Figs.
(PDF)

**S5 Table. Percentage of known SARS-CoV-2 genomic sequences with mutations in primer/ probe binding sites for RdRp, M, and S gene assays.** High-quality genome sequences analysed only ($n$ = 97,782). RdRp, RNA-dependent RNA polymerase; SARS-CoV-2, Severe Acute Respiratory Syndrome Coronavirus 2.
(PDF)

**S6 Table. Interpretation and suggested action based on N1E-RP or N2E-RP qRT-PCR results.** qRT-PCR, quantitative reverse transcription PCR.
(PDF)

**S1 Fig. RNA extraction has no substantial impact on the sensitivity of the N1E-RP and N2E-RP 4-plex assays.** N1E-RP and N2E-RP RT-qPCR assays were performed on (A) 1 to 10,000 copies of SARS-CoV-2 control RNA (IVT) before (as Fig 2A) and after nucleic acid extraction, (B) a serial dilution of RNA isolated from cultured SARS-CoV-2, before (as Fig 2B) and after re-extraction. Mean ± SD for technical triplicates; $R^2$ values for logarithmic trend line fitting and amplification efficiencies (E) for samples after (re)extraction. Also, see S1 and S2 Tables and S1 Data. IVT, in vitro transcribed; SARS-CoV-2, Severe Acute Respiratory Syndrome Coronavirus 2; SD, standard deviation.
(PDF)

**S2 Fig. High reproducibility for extraction controls, but high variability for the human *RPP30* control in NTS samples.** (A, B) Cq values for internal controls, MS2 for TaqPath and PhHV for N1E-RP and N2E-RP assays (A), and *RPP30* controls (B). (C) Cq values for PhHV and *RPP30* controls for N1E-RP and N2E-RP assays, ranked by *RPP30* values from the N1E-RP assay, confirm that variability does not substantially correlate with extraction efficiency. Also, see Table 1, S4 Table, and S1 Data. Cq, cycle quantification; NTS, nose and throat swabs; PhHV, Phocine Herpes Virus.
(PDF)

**S3 Fig. The human *RPP30* control probe can detect 10 copies of control DNA.** (A) Cq values for *RPP30* on a serial dilution of positive control plasmid DNA (100,000 down to 10 copies were tested). (B) Cq values for *RPP30* on NAs isolated from human cultured cells (1 = undiluted) and NA isolated from a serial dilution of the same cell suspension show a strong linear correlation and 92% amplification efficiency. Negative control samples did not show any amplification. Data points and error bars, mean ± SD ($n$ = 2 technical replicates). $R^2$ values for logarithmic trend line fitting; E, amplification efficiency. Also, see S1 Data. Cq, cycle quantification; NA, nucleic acid; SD, standard deviation.
(PDF)

**S4 Fig. Increased chance of false negatives for low quality NTS samples, with high *RPP30* Cq values.** (A, B) Comparison of actual (Act) and *RPP30*-normalised (Norm) Cq values for SARS-CoV-2 targets shows that values cluster more strongly after normalisation, both when normalising to the worst (highest *RPP30* Cq; A) and best (lowest *RPP30* Cq; B). Increased clustering (i.e., reduced variability) demonstrates the impact of the correlation between quantity of human and viral nucleic acids in NTS samples (Fig 4C). This shows (1) that sample quality substantially impacts on assay sensitivity; and (2) that variability in viral loads is smaller than non-normalised data suggest. Plots in (A) demonstrate that 4–6 positives (15%–23%) would likely have been below the detection limit (above the red line, viral Cq > 40) in a worst quality

sample scenario. Also, see Table 1, S4 Table, and S1 Data. Cq, cycle quantification; NTS, nose and throat swabs; SARS-CoV-2, Severe Acute Respiratory Syndrome Coronavirus 2
(PDF)

**S1 Protocol. Four-plex SARS-CoV-2 RT-qPCR assays.** RT-qPCR, quantitative reverse transcription PCR; SARS-CoV-2, Severe Acute Respiratory Syndrome Coronavirus 2.
(PDF)

**S2 Protocol. Viral nucleic acid isolation.**
(PDF)

**S1 File. qRT-PCR primers and probes on the SARS-CoV-2 genome, Wuhan-Hu-1 isolate (SnapGene file).** qRT-PCR, real-time quantitative reverse transcription PCR; SARS-CoV-2, Severe Acute Respiratory Syndrome Coronavirus 2.
(DNA)

**S1 Data. Source data for Table 2, Figs 1–4, S5 Table, S1–S4 Figs, and Fig P1.**
(XLSX)

## Acknowledgments

We thank Sylvie Behillil (Institut Pasteur, Paris, France), Rory Gunson (NHS Molecular Development in Virology and Microbiology, Glasgow, UK), and Paul Wallace (QCMD, Glasgow, UK) for reagents and information, and Angela Ingram, Derek Mills, Maggie Arbuckle, Joyce Begbie, Heather Coupar, Eilidh Guild, Samantha Griffiths, Garry Jempson, Alain Kemp, Frances Rae, Maggie MacDonald and Thomas Williams for support. This work was part of a technology development exercise at the NHS-led University of Edinburgh/NHS Lothian COVID-19 testing centre at the MRC Institute of Genetics and Molecular Medicine.

## Author Contributions

**Conceptualization:** Martin A. M. Reijns, Andrew P. Jackson.

**Data curation:** Martin A. M. Reijns.

**Formal analysis:** Martin A. M. Reijns, Louise Thompson, Holly A. Black, David A. Parry, Alan O'Callaghan.

**Funding acquisition:** Andrew P. Jackson.

**Investigation:** Martin A. M. Reijns, Louise Thompson, Juan Carlos Acosta, Holly A. Black, Alison Daniels, Maria C. Sanchez.

**Methodology:** Martin A. M. Reijns, Louise Thompson, Juan Carlos Acosta, Francisco J. Sanchez-Luque, Austin Diamond, David A. Parry, Alison Daniels, Carolina Uggenti, Maria C. Sanchez, Louise Slater, Morad Ansari, Kate Templeton, Jürgen G. Haas, Nick Gilbert, Ian R. Adams.

**Project administration:** Martin A. M. Reijns, David Moore, Ian R. Adams, Andrew P. Jackson.

**Resources:** Francisco J. Sanchez-Luque, Marie O'Shea, Carolina Uggenti, Michelle L. L. McNab, Martyna Adamowicz, Elias T. Friman, Toby Hurd, Edward J. Jarman, Frederic Li Mow Chee, Jacqueline K. Rainger, Marion Walker, Camilla Drake, Dasa Longman, Christine Mordstein, Sophie J. Warlow, Stewart McKay, Ian P. M. Tomlinson, David Moore,

Nadine Wilkinson, Jill Shepherd, Kate Templeton, Ingolfur Johannessen, Christine Tait-Burkard, Jürgen G. Haas, Ian R. Adams.

**Software:** David A. Parry.

**Supervision:** Martin A. M. Reijns, Austin Diamond, Kate Templeton, Christine Tait-Burkard, Jürgen G. Haas, Ian R. Adams, Andrew P. Jackson.

**Visualization:** Martin A. M. Reijns.

**Writing – original draft:** Martin A. M. Reijns.

**Writing – review & editing:** Martin A. M. Reijns, Juan Carlos Acosta, Francisco J. Sanchez-Luque, Jürgen G. Haas, Nick Gilbert, Ian R. Adams, Andrew P. Jackson.

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
