## [Editor Report · Decision Letter 0]

10 Nov 2020

Dear Dr. Reijns, 

Thank you for submitting your manuscript entitled "A sensitive and affordable multiplex RT-qPCR assay for SARS-CoV-2 detection" for consideration as a Methods and Resources by PLOS Biology.

Your manuscript has now been evaluated by the PLOS Biology editorial staff as well as by an academic editor with relevant expertise and I am writing to let you know that we would like to move forward with your manuscript.

However, before we can send you the official Minor decision letter, we need you to complete your submission by providing the metadata that is required for full assessment. To this end, please login to Editorial Manager where you will find the paper in the 'Submissions Needing Revisions' folder on your homepage. Please click 'Revise Submission' from the Action Links and complete all additional questions in the submission questionnaire.

Also, as part of a commitment between PLOS (and other publishers) and the World Health Organisation to ensure that all relevant clinical information about this outbreak is shared quickly, we will be notifying the WHO directly about your study/manuscript. We strongly encourage you to post your manuscript as a preprint if you have not already done so. If your article is in scope for biorxiv we can assist with this for you. Please see here for the statement coordinated by the Wellcome Trust: https://wellcome.ac.uk/press-release/sharing-research-data-and-findings-relevant-novel-coronavirus-covid-19-outbreak.

Please re-submit your manuscript within two working days, i.e. by Nov 12 2020 11:59PM.

Kind regards,

Paula

---

Associate Editor

PLOS Biology

---

## [Editor Report · Decision Letter 1]

11 Nov 2020

Dear Dr. Reijns,

Thank you very much for submitting your manuscript "A sensitive and affordable multiplex RT-qPCR assay for SARS-CoV-2 detection" for consideration as a Methods and Resources by PLOS Biology. Since this is a Review Commons manuscript, it was evaluated by the PLOS Biology editors as well as by an Academic Editor with relevant expertise. We all appreciated the attention to an important topic. 

We're delighted to let you know that we're now editorially satisfied with your manuscript. However before we can formally accept your paper and consider it "in press", we also need to ensure that your article conforms to our guidelines. A member of our team will be in touch shortly with a set of requests. As we can't proceed until these requirements are met, your swift response will help prevent delays to publication. Please also make sure to address the data and other policy-related requests noted at the end of this email.

- a cover letter that should detail your responses to any editorial requests, if applicable

*Copyediting*

*Published Peer Review History*

*Early Version*

Sincerely,

Paula

---

Associate Editor,

pjaureguionieva@plos.org,

PLOS Biology

DATA POLICY:

Table 2, Table S5, Figure 1D, 2C, 3, 4, S2, S3, S4.

Please also change the name of the figure in the supplementary protocol 2 since now has the same name as supplementary figure 4, and please supply data for it as well.

---

## [Editor Report · Decision Letter 2]

23 Nov 2020

Dear Dr Reijns,

On behalf of my colleagues and the Academic Editor, Bill Sugden, I am pleased to inform you that we will be delighted to publish your Methods and Resources in PLOS Biology. 

PRODUCTION PROCESS

Before publication you will see the copyedited word document (within 5 business days) and a PDF proof shortly after that. The copyeditor will be in touch shortly before sending you the copyedited Word document. We will make some revisions at copyediting stage to conform to our general style, and for clarification. When you receive this version you should check and revise it very carefully, including figures, tables, references, and supporting information, because corrections at the next stage (proofs) will be strictly limited to (1) errors in author names or affiliations, (2) errors of scientific fact that would cause misunderstandings to readers, and (3) printer's (introduced) errors. Please return the copyedited file within 2 business days in order to ensure timely delivery of the PDF proof. 

If you are likely to be away when either this document or the proof is sent, please ensure we have contact information of a second person, as we will need you to respond quickly at each point. Given the disruptions resulting from the ongoing COVID-19 pandemic, there may be delays in the production process. We apologise in advance for any inconvenience caused and will do our best to minimize impact as far as possible.

EARLY VERSION

PRESS 

Kind regards,

Alice Musson, 

PLOS Biology

on behalf of

Editor

PLOS Biology